# Tidal Resonance: A Factor Worth Considering in the Orbital Evolution of Heartbeat Stars

Jian-Wen Ou [1,*], Chen Jiang [2], Ming Yang [3], Cong Yu [4,5], Dong-Yang Gao [6] and Guangbo Long [1]

1 School of Intelligent Engineering, Shaoguan University, Shaoguan 512005, China; longgb@mail2.sysu.edu.cn
2 Max-Planck-Institut für Sonnensystemforschung, Justus-von-Liebig-Weg 3, 37077 Göttingen, Germany; jiangc@mps.mpg.de
3 College of Surveying and Geo-Informatics, Tongji University, Shanghai 200092, China; myang@tongji.edu.cn
4 School of Physics and Astronomy, CSST Science Center for the Guangdong-Hongkong-Macau Greater Bay Area, Sun Yat-sen University, Zhuhai 519082, China; yucong@mail.sysu.edu.cn
5 State Key Laboratory of Lunar and Planetary Sciences, Macau University of Science and Technology, Macau 999078, China
6 Shandong Key Laboratory of Optical Astronomy and Solar-Terrestrial Environment, Institute of Space Sciences, School of Space Science and Physics, Shandong University, Weihai 264209, China; gaodongyang@sdu.edu.cn
* Correspondence: o_jw@163.com

**Abstract:** Heartbeat star systems have been reported to exhibit two distinct different orbital dynamic evolution processes: apsidal precession (e.g., KIC 4544587) and orbital decay (e.g., KIC 3766353). While experiencing similar dynamic tidal interactions, these binary systems display different dynamical behaviors, which is a puzzling phenomenon. In this work, we deduced a theoretical relation between the timescale of stellar pulsation $P_{pul}$ and orbital periods $P_{orb}$ of heartbeat stars based on the resonance criteria representing the orbital local low-energy configuration. The theoretical relation shows that when the ratio of $P_{orb}$ to $P_{pul}$ is an integer, the specific orbital period is captured in the resonance state, resulting in resonance locking. The resonance criteria are verified by periodograms of the pulsations and orbits of the two systems KIC 4544587 and KIC 3766353 from observations. KIC 4544587 is an apsidal precession heartbeat star with eight observed resonant frequencies available from observations and has an almost integer ratio of $P_{pul}/P_{orb} = 67.968$. On the contrary, KIC 3766353 is undergoing the process of orbital shrinkage with only three weak pulsation–orbital resonance frequencies available and shows a non-integer ratio of $P_{pul}/P_{orb} = 83.163$. Given the results, the theoretical relation is a potential proxy to distinguish between apsidal precession and orbital decay binary systems. Furthermore, we predict that the orbital period of KIC 3766353 will be reduced to 2.492 days, at which time it will be transformed into apsidal precession.

**Keywords:** eclipsing binaries; stellar oscillations; tidal resonance; KIC 3766353; KIC 4544587

## 1. Introduction

Heartbeat (HB) stars are highly eccentric (with eccentricity $e \gtrsim 0.2$) binary star systems with pulsations caused by tidal interactions, whose light curves significantly brighten when near the periastron due to tidal distortion, reflection, and Doppler beaming [1–3]. The shape of the light curve is similar to that of an electrocardiogram of a human heart, which is why this type of binary star is called a heartbeat star..

In an HB star, due to the companion moving close to and away from the primary, periodic gravitational forces induce the dynamic tides that can be revealed as tidally excited oscillations (TEOs) in the observed flux [3,4]. Typically, HB stars are observed in systems with orbital periods ranging from a fraction of a day to tens of days. It was not until the *Kepler* space telescope era that HB stars were widely studied because of the continuous and high-precision photometry data needed to unambiguously identify the tiny dynamic tides of typical relative flux variations of $\sim 10^{-3}$. For instance, the prototype of HB stars, KOI-54

(HD 187091), was discovered by *Kepler* and has been extensively studied in numerous works [5–8].

HB stars are expected to be easily observed in red giants, because red giants are susceptible to the tidal effects of their companions due to the large radii and low surface gravities (log *g*). A sketchy analysis of HB stars containing red giant stars was studied in the work of Beck et al. [9]. Thanks to the ground-based spectroscopy and *Kepler* photometry, another set of HB stars consisting of stars located on or close to the main sequence was found, including seventeen systems analyzed by Thompson et al. [1], 22 by Shporer et al. [2], and seven by Dimitrov, Kjurkchieva, & Iliev [10]. Those HB star systems are mainly low- and intermediate-mass A-type to F-type stars.

The periodically changing gravitational effect that causes stellar pulsation affects the orbital dynamic configuration in binary systems via tidal dissipation [11–13], and more new mechanisms of tidal dissipation are advocated to explain realistic dynamical studies in stellar populations or star–planet binary systems [14,15]. We know that dissipative forces are driving evolutionary processes in our solar system [16]. Outside the solar system, the observed sample of HB stars exhibiting TEOs may be a valuable test bed for studies on the influence of dynamical tides on orbital evolution in binary systems.

There are several reasons why the orbital period of an HB star might change. An eccentric orbit will undergo apsidal precession due to general relativity and the Newtonian correction from rotational and tidal deformation [17,18]. There are also the long-term effects of tidal dissipation, which, for highly eccentric systems, are expected to lead to orbital coplanarization, circularization, and decay [19]. The most interesting are probably apsidal precession and tidal orbital decay, because the measured rate of orbital period change would give us insight into a poorly understood phenomenon. But, the study of dynamical tides during orbital evolution has been a challenge to carry out due to the limited number of HB stars with detailed quantitative parameters. To investigate the subtle relationship between TEOs and orbital evolution, in this work, we focused on two binary star systems: KIC 4544587 and KIC 3766353. These systems feature detailed stellar parameters but exhibit different orbital dynamics.

The structure of this paper is as follows: In Section 2, we describe the samples of HB stars with both primary and secondary companion masses and give a brief overview of two HB star systems with completely different dynamical behaviors, i.e., KIC 4544587 with apsidal precession and KIC 3766353 with orbital decay. In Section 3, we present our analysis of the two HB stars with *Kepler* data. In Section 4, we describe the pulsation–orbital resonance states of two HB stars from the perspective of light curve periodograms. In Section 5, we deduce a simple potential criterion to determine whether the pulsation–orbital periods of HB stars are resonant. The orbital evolution of KIC 3766353 is presented in Section 6, where we predict the evolution of HB star systems from orbital decay to orbital precession. Finally, we discuss the possible implications of the observed pulsation–orbital resonance to improve our understanding of apsidal precession and orbital decay in Section 7.

## 2. Samples of HB Stars

Stellar mass and effective temperature are the fundamental parameters of stars and can be used to estimate the evolutionary state of each companion. The components of KIC 4544587 are both massive A-type stars, while the components of KIC 3766353 are much cooler and less massive stars, i.e., F–K stars. While the fundamental parameters of these two HB star systems differ significantly, the collected stellar masses of HB stars, as shown in Figure 1, demonstrate that their components cover a wide range of masses, including those of KIC 4544587 and KIC 3766353. The specific parameters of these two targets are shown in Table 1.

**Table 1.** The orbital and stellar parameters for KIC 4544587 and KIC 3766353, obtained from Table 6 of Hambleton et al. [20] and Table 1 of Ou et al. [21], respectively.

| Orbital Parameters | KIC 4544587 | | KIC 3766353 | |
|---|---|---|---|---|
| Dynamic type: | apsidal precession (mrad yr$^{-1}$): $42.97 \pm 0.18$ * | | orbital decay (ms yr$^{-1}$): $358.04^{+45.74}_{-53.72}$ | |
| Orbital period (d): $P_{\text{orb}}$ | $2.1890951 \pm 0.0000007$ | | $2.666985 \pm 0.000004$ | |
| Semi-major axis ($R_\odot$): $a$ | $10.855 \pm 0.046$ | | $8.38^{+0.68}_{-0.21}$ | |
| Orbital eccentricity: $e$ | $0.275 \pm 0.004$ | | $0.264 \pm 0.004$ | |
| Argument of periastron (°): $\omega$ | $5.74 \pm 0.03$ | | $-6.40^{+8.72}_{-1.12}$ | |
| Orbital inclination (°): $i$ | $87.9 \pm 0.03$ | | $78.89^{+0.53}_{-0.63}$ | |
| Stellar parameters | Primary | Secondary | Primary | Secondary |
| Mass ($M_\odot$) | $1.98 \pm 0.07$ | $1.61 \pm 0.06$ | $0.76^{+0.21}_{-0.13}$ | $0.35^{+0.13}_{-0.16}$ |
| Radius ($R_\odot$) | $1.82 \pm 0.03$ | $1.58 \pm 0.03$ | $1.32^{+0.16}_{-0.06}$ | $0.34^{+0.04}_{-0.03}$ |
| $T_{\text{eff}}$ (K) | $8600 \pm 100$ | $7750 \pm 180$ | $6757^{+178}_{-392}$ | $4381^{+175}_{-164}$ |
| log $g$ (cgs) | $4.241 \pm 0.009$ | $4.33 \pm 0.01$ | $4.08^{+0.22}_{-0.11}$ | $4.92^{+0.26}_{-0.27}$ |

*Note*: * is the latest measured value from the reference [22].

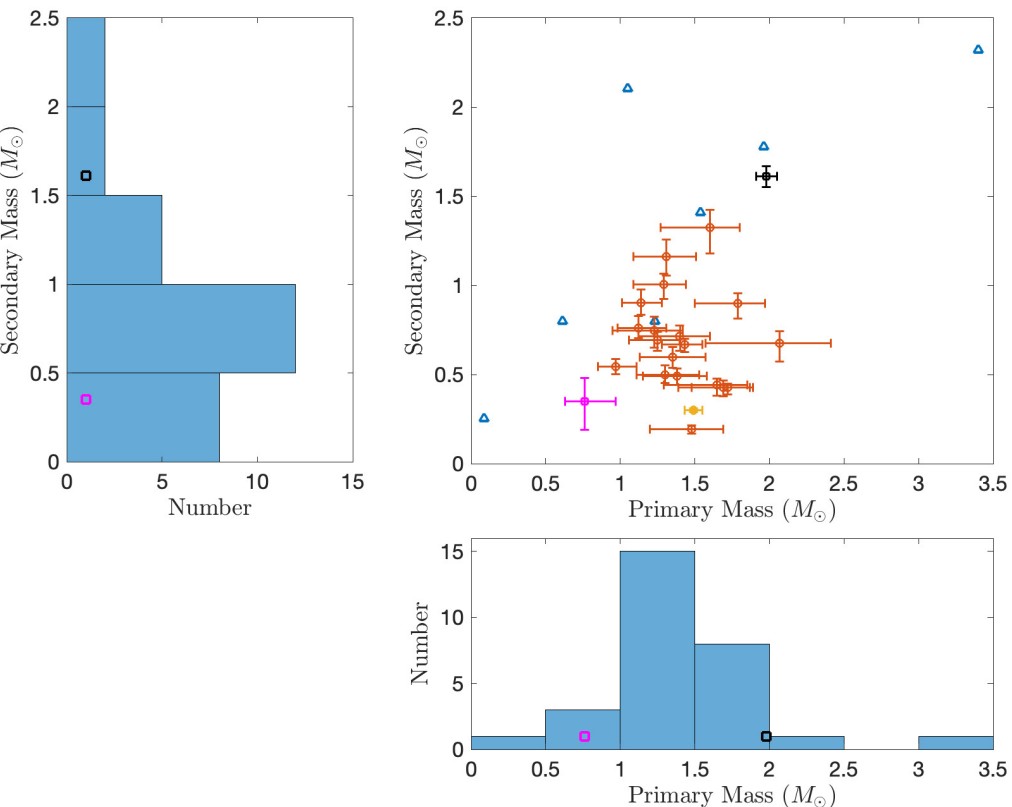

**Figure 1.** Mass distribution of the primary and secondary companions of HB stars. Orange and yellow cycles indicate nineteen HB stars studied by Shporer et al. [2] and one HB star studied by Beck et al. [9] with error bars; blue triangles indicate seven HB stars studied by Dimitrov, Kjurkchieva, & Iliev [10] but without uncertainties. The black and magenta squares indicate KIC 4544587 and KIC 3766353, respectively.

HB stars are divided according to the geometry of the light curve, which is closely related to the orbital configuration characteristics of the binary system, such as high

eccentricity and the orbital inclination [23]. Figure 2 shows the time-domain data for *Kepler* quarter (Q) 9 for KIC 4544587 and KIC 3766353 at the same observation time. The shapes of the light curves of these two targets are very similar, except for the different depths of binary eclipsing caused by the different orbital inclinations. For comparison purposes, the time-domain data are converted to phase data in the bottom panel of Figure 2. The magnified image of the out-of-eclipse part shows a remarkable heartbeat signature at the phases of ∼0.78 and ∼0.83 for KIC 4544587 and KIC 3766353, respectively. And, TEOs (i.e., the fluctuations in the light curve) throughout the orbital phase are clearly visible in the graph.

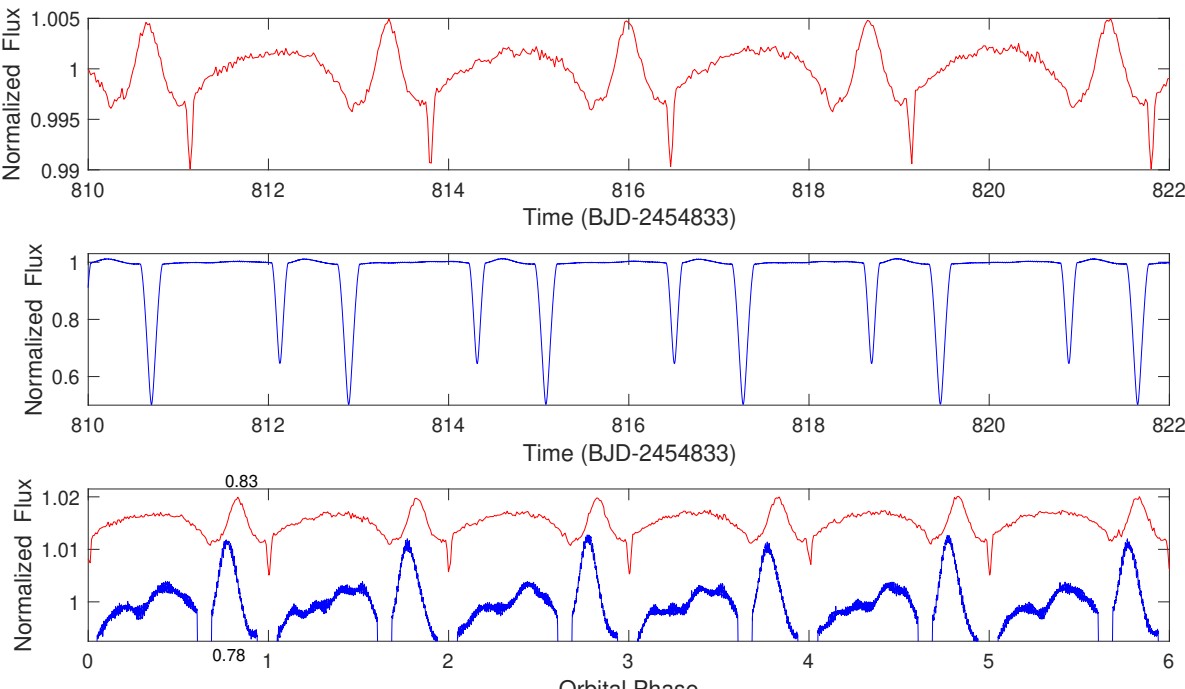

**Figure 2.** The normalized fluxes of *Kepler* Q9 data for KIC 3766353 (top panel) and KIC 4544587 (middle panel) at the same observation time are marked in red and blue, respectively. The normalized flux is derived from the average value of the out-of-eclipse parts. The bottom panel is a magnified image of the out-of-eclipse part whose abscissa axis is in the orbital phase, and the light curves of KIC 3766353 are offset by 0.015 relative flux units for clarity. The shapes of the light curve are almost the same.

The eclipse timing variations (ETVs) represent a traditional approach to the investigation of orbital period changes. For KIC 4544587, the apsidal precession is $42.97 \pm 0.18$ mrad yr$^{-1}$, as measured by eclipse timing variations [22]. Ou et al. quantitatively measured the contribution of the dynamic tides to the apsidal precession as $19.05 \pm 1.70$ mrad yr$^{-1}$ [22], resolving the long-standing controversy regarding the contribution of dynamic tides to apsidal precession from the observations [24–27].

KIC 3766353 is one of the HB stars with TEOs. Almost simultaneously, Ou et al. [21] measured the primary and secondary ETVs. The timing analysis shows that KIC 3766353 is a hierarchical triple system with a hidden third body in its outer orbit and a red dwarf in its inner orbit. The period decay rate of the inner orbit is approximately 358 ms yr$^{-1}$. In theory, tidal dissipation of the primary star caused by dynamic tides effectively transfers the orbital angular momentum of the binary system, thus causing the period decay of the binary orbit [21].

### 3. *Kepler* Data and Their Analysis

*3.1. Photometric Data and Data Preprocessing*

KIC 4544587 has a *Kepler* magnitude $K_p = 10.801$. The photometric data include long-cadence data (29.4 min) during Q0–17 and short-cadence data during Q3.2 and 7–10. We only used the *Kepler* short-cadence (58.89 s) data during Q7–10, which were enough to meet the precision requirements of our analysis.

KIC 3766353 has a *Kepler* magnitude $K_p = 13.968$, which is much fainter than that of KIC 4544587. Therefore, short-cadence data are not available for KIC 3766353. The data used in this paper are the entire photometric data. In total, 13 quarters of *Kepler* data were collected, because the target was not observed in Q0, Q6, Q10, or Q14.

In order to remove the long-term systematic trend due to the drift of the space telescope, we detrended and normalized the original light curve by fitting a low-order ($\leq$6th) polynomial to the individual segments of each quarter. Only the out-of-eclipse light curve was selected to fit the polynomials, and the specific order was chosen as the one that minimized the standard deviation. The outliers of all data points were clipped by a $5\sigma$ criterion.

*3.2. Binary Model and Frequency Extraction*

Eclipsing binary systems with pulsation brings both opportunities and challenges to the study of stellar physics. On one hand, stellar parameters such as the mass, radius, and age can be obtained through asteroseismic analysis. Then, the orbital parameters can be restricted to acquire the orbital structure and evolutionary process of the binary system. However, on the other hand, the pulsation signal interferes with the orbital signal analysis, and vice versa, the orbital signal interferes with the pulsation extraction. In order to extract the pulsation signals from the star, the orbital signals from the eclipsing binary must be eliminated first. The binary modeling package, PHysics Of Eclipsing BinariEs (PHOEBE; Prša et al. [28], Conroy et al. [29]), is used to model the exotica binary systems in this paper.

When modeling the light curves, the initial inputs are taken from Hambleton et al. [20] and Ou et al. [21] for KIC 4544587 and KIC 3766353 (can be found in Table 1), respectively. Once the initial model had been generated, the differential corrections algorithm was applied in an iterative process to obtain an accurate fit for the light curve data. Using the results of previous studies [20–22], we reproduced the binary model light curve.

Considering the unequal gaps in the observational data, a convenient approximation of the Fourier transform is the Lomb–Scargle periodogram. We used a standard Lomb–Scargle periodogram to detect periodic signals in the light curves [30–32]. A simple frequency uncertainty can be derived using the theoretical estimation presented by Kallinger, Reegen, & Weiss [33],

$$\sigma(f) = \sqrt{\frac{6}{N}} \frac{1}{\pi T} \frac{\sigma(m)}{A},$$ (1)

where $N$ is the total measurement of the data set, $T$ is the full time baseline, $\sigma(m)$ is the root-mean-square deviation of the noise in the observed magnitudes, and $A$ is the amplitude. In this paper, we chose the Monte Carlo method to estimate the uncertainty in the frequency and amplitude as follows: 10,000 simulated light curves were generated with each data point following a normal distribution defined by its original observation value and error. The periodogram was repeated for each simulated light curve to extract the frequencies and amplitudes of the pulsation modes. Finally, $1\sigma$ uncertainties were obtained from the frequency and amplitude distributions.

We statistically evaluated the evidence for a signature with the false alarm probability (FAP). The convention defined by Donati et al. [34] was adopted so that a pulsation was unambiguously detected whenever the associated probability was larger than 99.9%, i.e., a FAP of smaller than 0.1%. We extracted pulsations with a signal-to-noise ratio SNR > 4. The signal is defined as the amplitude of the peak frequency, and the noise is defined as the mean amplitude of the background spectrum in a window size of 0.1 d$^{-1}$ without the

central peak. We were interested in frequencies that were in resonance with the orbital period. Finally, the frequencies near the resonant frequencies were extracted. All extracted frequencies are listed in Table 2.

**Table 2.** The identified TEOs from the apsidal precession system KIC 4544587. The values in parentheses give the $1\sigma$ uncertainty for the previous digit.

| Frequency (d$^{-1}$) | Amplitude (Relative) | Comment ($P_{\text{orb}} = 2.189$ d) |
|---|---|---|
| 0.45663 (1) | 1.000 (1) | $f_{\text{orb}}$ |
| 0.91393 (2) | 0.288 (2) | $2f_{\text{orb}}$ |
| 1.37055 (1) | 0.800 (2) | $3f_{\text{orb}}$ |
| 1.61170 (10) | 0.003 (1) | |
| 1.82718 (2) | 0.128 (2) | $4f_{\text{orb}}$ |
| 2.01293 (2) | 0.043 (1) | |
| 3.19773 (3) | 0.028 (1) | $7f_{\text{orb}}$ |
| 3.46860 (5) | 0.017 (1) | |
| 3.65435 (3) | 0.037 (1) | $8f_{\text{orb}}$ |
| 4.11098 (5) | 0.016 (2) | $9f_{\text{orb}}$ |
| 4.56828 (6) | 0.012 (2) | $10f_{\text{orb}}$ |

## 4. Observation of the Tidal Resonance State

### 4.1. KIC 4544587 with Pulsation–Orbital Resonance

The periodograms of the binary orbital period and the stellar pulsation are shown in the top and bottom panels of Figure 3, respectively. The black line in the top panel is a periodogram of the whole light curve containing orbital eclipse information. In the bottom panel, the blue line is the periodogram constructed by only the out-of-eclipse part of the residual after subtracting the orbit of the binary model produced by PHOEBE, as mentioned in Section 3.2. The vertical blue dotted lines and the numbers above them show the multiple periods of the orbit (i.e., $1/n$)[1]. The vertical cyan lines mark the resonant TEO frequencies and the width of the cyan line indicates the spectral resolution. It can be seen from Figure 3 that the pulsation frequency of KIC 4544587 is exactly 1–10 times the orbital frequency but is missing the resonant frequencies $n = 5$ and 6. The reason for the lack of the resonant $n = 5$ and 6 pulsations may be related to the inner structure of the primary star, which is beyond the scope of this paper. The eight resonant TEOs were identified in KIC 4544587 [20], as detailed in Table 2.

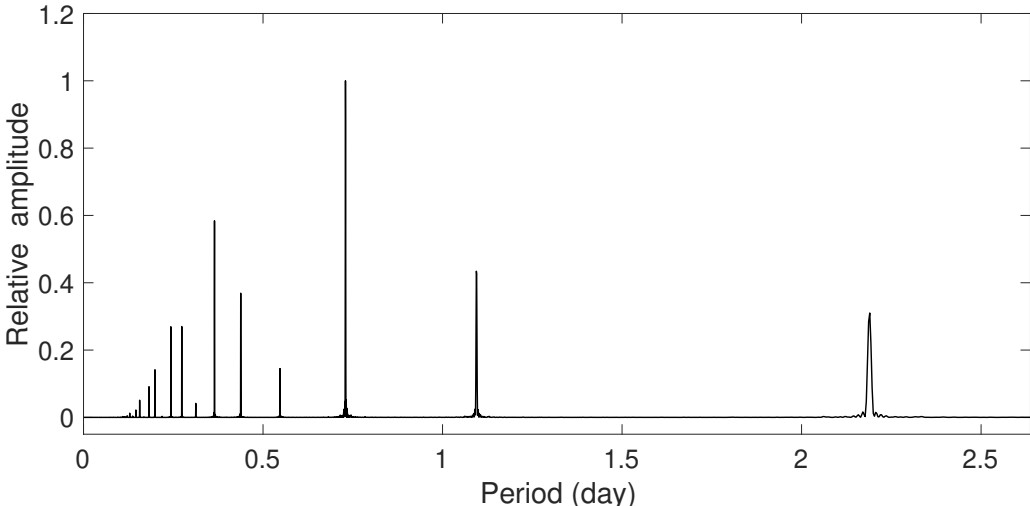

**Figure 3.** *Cont.*

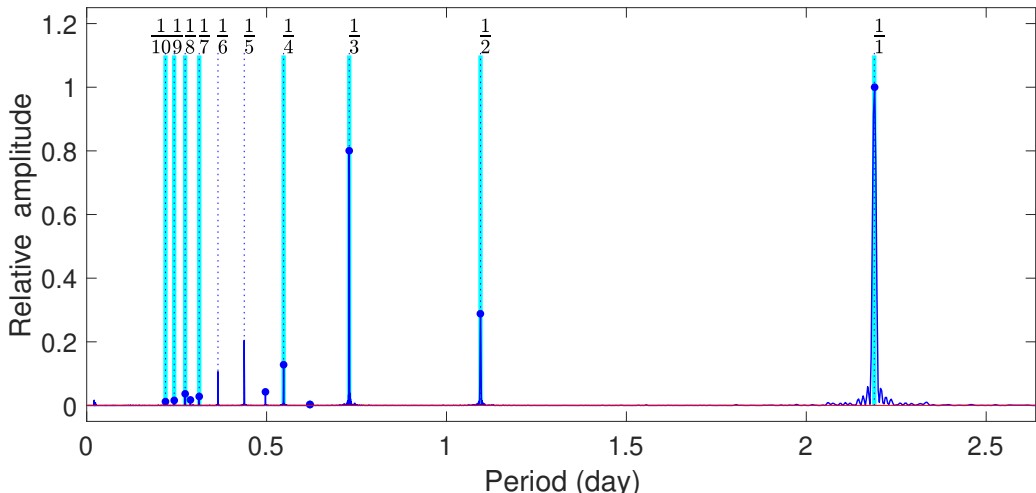

**Figure 3.** Periodograms for KIC 4544587. Top panel: the black line is the spectrum of the light curve that contains the orbital information. Bottom panel: the blue line is the spectrum of the pulsation after removing the orbital information, and the blue dots mark the extracted pulsation frequencies. The thin horizontal red line indicates the 0.1% false alarm probability. The vertical cyan lines denote the resonance frequencies within the Rayleigh resolution, and the inverse of the numbers above them represent the number of times they are the orbital frequency. Eight resonant peaks are observed for KIC 4544587, which is an apsidal precession HB star.

### 4.2. KIC 3766353 without Pulsation–Orbital Resonance

Again, using the data analysis method described in Section 3.2, we examined the frequencies of stellar pulsation and orbital signals. From Figure 4, it is clear that only three pulsation frequencies are multiples of the orbital frequency 0.3751 d$^{-1}$ (corresponding orbital period $P_{\mathrm{orb}} = 2.666$ d) within the error, that is, TEOs $n = 3, 5$, and 8. The details are tabulated in Table 3.

In addition to the frequency, the amplitudes of the TEOs should be noteworthy. In a binary system with an eccentric orbit, when the orbital frequency is close to the integer harmonics of a stellar intrinsic frequency, the amplitudes of TEOs increase via resonances between the orbital harmonics and mode frequencies [3,4]. But, in Figure 4, we do not see a significant increase in the amplitudes for TEOs $n = 3, 5$, and 8. This orbital state implies that the current orbital period $P_{\mathrm{orb}} = 2.666$ d does not resonate with the stellar eigenfrequency. In other words, the binary system does not reach a stable state with a lower orbital energy.

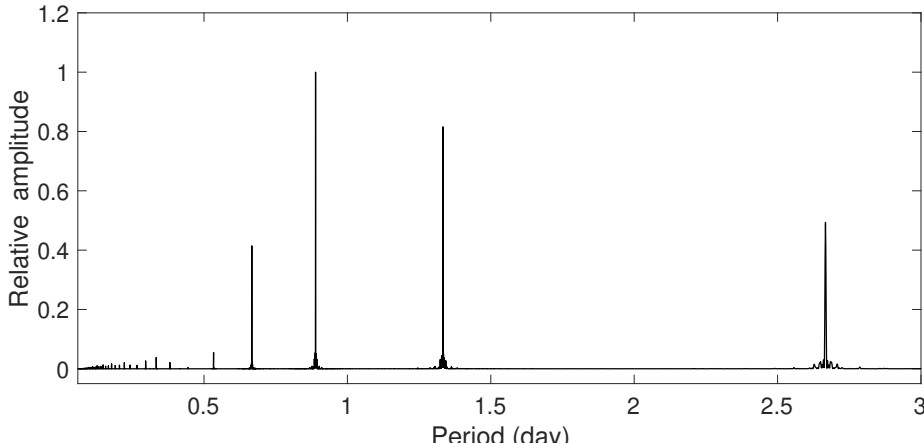

**Figure 4.** *Cont.*

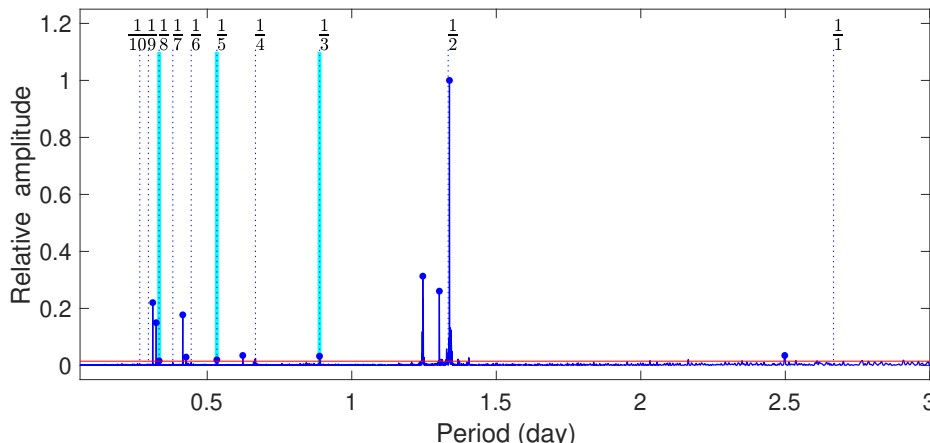

**Figure 4.** The same as Figure 3 but for KIC 3766353. An orbital decay HB star with only three resonant TEOs: $n = 3, 5$, and 8.

**Table 3.** The identified pulsation frequencies of orbital decay system KIC 3766353. The last column is the predicted state with a short orbital period of 2.492 days that achieves resonance with four higher amplitude pulsation frequencies.

| Frequency (d$^{-1}$) | Amplitude (Relative) | Comment ($P_{orb} = 2.666$ d) $^\dagger$ | Comment ($P'_{orb} = 2.492$ d) $^\S$ |
|---|---|---|---|
| 0.40031 (5) | 0.035 (4) | | |
| 0.74735 (1) | 1.000 (2) | | |
| 0.76739 (1) | 0.260 (2) | | |
| 0.80268 (1) | 0.313 (2) | | $2f'_{orb}$ |
| 1.12557 (3) | 0.032 (3) | $3f_{orb}$ | |
| 1.60536 (3) | 0.035 (2) | | $4f'_{orb}$ |
| 1.87531 (5) | 0.020 (2) | $5f_{orb}$ | |
| 2.34329 (4) | 0.029 (2) | | |
| 2.40786 (1) | 0.177 (2) | | $6f'_{orb}$ |
| 2.99711 (7) | 0.016 (4) | $8f_{orb}$ | |
| 3.09081 (1) | 0.150 (2) | | |
| 3.21054 (1) | 0.220 (2) | | $8f'_{orb}$ |

$^\dagger$: The current state is orbital decay, and the three resonant frequencies have lower amplitudes. $^\S$: The prediction state is apsidal precession, and the four resonant frequencies have higher amplitudes.

## 5. Two Different Orbital Dynamics

### 5.1. A Potential Simple Relationship between Pulsation and the Orbital Period

Stable configurations in binaries are usually connected with the origins of some resonances [19,35,36]. Tidal resonance causes the enhancement of the interaction between celestial bodies and then has a great influence on the stability of celestial bodies' motion. A resonance can arise when the rotational and orbital periods (or frequencies) of two or more bodies satisfy a simple numerical relationship. However, more complicated resonant relationships also exist for $N$-body systems, for instance, our solar system [16].

From an orbital energy point of view, binary stars tend to move the system to a lower energy configuration. The lower-energy configurations, including orbital circularization [14,37,38], the period of spin-orbit synchronization [35,39,40], and the axis of spin-orbit alignment [41–44], have been investigated in many studies in the literature. In fact, the spin-orbit synchronization is a 1:1 resonance between the stellar spin period and the orbital period. Additionally, the orbital local lower-energy configuration caused by passage through a pulsation–orbital resonance and resonant locking is also worth considering [36,45–47].

Consider a primary star with a pulsation period of $P_{\text{pul}}$ and an orbital period of $P_{\text{orb}}$ in a HB star system. The pulsation–orbital resonance refers to the harmonic of their period, as

$$P_{\text{n}} = \frac{P_{\text{orb}}}{P_{\text{pul}}} \, , \tag{2}$$

where $P_{\text{n}}$ is an integer or a simple numerical ratio.

The orbital period $P_{\text{orb}}$ of the binary system depends on the mass of the primary component $M_1$ as well as that of the secondary $M_2$ and the semi-major axis $a$ of the orbit and is defined based on the Kepler's third law

$$P_{\text{orb}} = 2\pi \sqrt{\frac{a^3}{G(M_1 + M_2)}} \, , \tag{3}$$

where $G$ is the gravitational constant. Subscripts "1" and "2" indicate the primary and secondary stars, respectively, in which the pulsating star is regarded as the primary.

For the pulsation period $P_{\text{pul}}$, the intrinsic property of the primary star immediately gives a dynamical timescale of the stellar oscillations. Indeed, the dynamical time scale can be regarded as a characteristic period of radial oscillations. It was realized very early on [48] that the period of stellar pulsation is approximately given by [49]:

$$P_{\text{pul}} = \sqrt{\frac{R_1^3}{GM_1}} \, , \tag{4}$$

where $R_1$ is the radius of the primary star.

Thus, we establish a link between the stellar pulsations and orbital periods, which is used to parameterize the stability of orbital evaluations,

$$P_{\text{n}} = \frac{P_{\text{orb}}}{P_{\text{pul}}} = 2\pi \sqrt{\frac{a^3}{(1 + M_2/M_1)R_1^3}} \, . \tag{5}$$

The resonance of the pulsation and orbit occurs when $P_{\text{n}}$ is an integer or a simple numerical ratio. Generally, $P_{\text{n}}$ is hardly an exact integer. We use the relative value of the rounding function to express the level of distance from the integer,

$$R_{\text{n}} = \frac{|\text{round}(P_{\text{n}}) - P_{\text{n}}|}{\text{round}(P_{\text{n}})} \times 100\% \, . \tag{6}$$

This relationship confirms a physical connection between the pulsation and orbital periods of HB star systems.

*5.2. Resonance Criterion of KIC 4544587 and KIC 3766353*

The resonance of stellar pulsation and the orbital period is an important factor affecting orbital evolution in HB stars that deserves attention. As mentioned in Section 5.1, the resonance state can be determined according to the value of $P_{\text{n}}$. Now, we continue to test this resonance criterion with data from KIC 4544587 and KIC 3766353.

By substituting the absolute values of $M_1$, $M_2$, $R_1$, and $a$ of KIC 4544587 and KIC 3766353 into Equation (5), we obtained the ratios of $P_{\text{orb}}$ to $P_{\text{pul}}$ as 67.968 and 83.163, respectively. The $P_{\text{n}}$ values are very high, in the order of $\sim$68 and $\sim$83, while none of the TEO frequencies presented in Tables 2 and 3 exceed $10f_{\text{orb}}$. The amplitude, or the detection ability, of the TEOs is determined by the Hansen coefficient $F_{nm}$ (details in Section 6, i.e., Equation (7)). The $F_{nm}$ coefficient expresses an intuitive principle, which states that the lower the multiple of the orbital frequency $n$, the higher the detection ability expected to be observed in TEOs.

For KIC 4544587, $P_{\text{n}}$ is very close to an integer. The relative value to an integer $R_{\text{n}}$ is 0.05%, which implies that the orbit and pulsation are in tidal resonance. The HB star system

of KIC 4544587 arrives at a stable configuration with lower orbital energy. On the contrary, $R_n$ of KIC 3766353 deviates from an integer with an $R_n$ value of about 0.2%. Therefore, KIC 3766353 has not yet reached a resonance state and still has a high orbital energy. In this case, the orbital angular momentum is easily and effectively transferred by tidally excited oscillations, resulting in orbital contraction.

Since the mass and radius of the HB star are almost impossible to determine precisely, their errors are relatively large, and taking the errors into account will make the $P_n$ uncertainty appear unreasonable. In other words, while the orbital and pulsation resonance state can be deduced using Equation (5) with the true values of $M_1, R_1, M_2$, and $a$, we may not be able to obtain $P_n$ as an integer from the observation, because $P_n$ is greatly affected by the error in the parameter. From Equation (4), the uncertainty of $P_{pul}$ given by the current observation data is shown to be $\sim$0.001 d and $\sim$0.01 d for KIC 4544587 and KIC 3766353, respectively, which is already very small. But, compared with the uncertainty of $P_{orb}$ obtained from the analysis of the light curve as $\sim$0.0000007 d and $\sim$0.000004 d, $P_{pul}$ is still very large, and Equation (2) will inevitably produce a large uncertainty.

In summary, the resonance criterion presented here is based on a very simple relationship between stellar pulsation and orbital evolution and it holds valid for the cases of KIC 4544587 and KIC 3766353. However, further observations of HB stars with various orbital dynamic evolution processes are necessary to confirm the validity of the criterion. In that case, the criterion could potentially provide insights into orbital and pulsating resonances, making it a valuable consideration to the study of the orbital evolution of HB stars.

## 6. Evolutions from Orbital Decay to Apsidal Precession

The mass $M_1$ and radius $R_1$ of the primary star determine the dynamic timescale of stellar pulsation $P_{pul}$, and the masses of the components $M_1, M_2$ and the orbital semi-major axis $a$ uniquely determine the orbital period $P_{orb}$. As discussed in Section 5.2, the ratio $P_n$ significantly deviates from an integer for KIC 3766353, which means that orbits in this configuration are not stable. The orbital period of KIC 3766353 is likely to shrink to a locally stable state with a lower orbital energy, e.g., resonance, through angular momentum and transfer mechanisms such as tidal dissipation, resonance capture, and mass transfer [15,50,51].

The orbital decay time scale of KIC3766353 is $\sim$0.6 million years [21]. Low-mass dwarfs have very long lifetimes, assuming that the time scale of pulsation is greater than the time scale of orbital decay. The angular momentum transfer in binary systems leads to a change in the orbital separation as the orbit evolves, resulting in a decrease in $P_{orb}$ (or an increase in the orbital frequency $f_{orb}$). The orbital frequency increases gradually and scans the eigenfrequencies of the primary star and finally arrives at integer harmonics of the stellar intrinsic frequencies. We manually increased the value of $f_{orb}$ from its current value of 0.3751 to 3.0 d$^{-1}$ with a step in the Rayleigh criterion of $1/\Delta T = 0.00068$ d$^{-1}$, where $\Delta T$ is the total observation duration of KIC 3766353. During this process, we searched for the $f_{orb}$ that resonated with the most stellar pulsation frequencies. Finally, we found that , in the best case of $f'_{orb} = 0.4013$ d$^{-1}$ or $P'_{orb} = 2.491716$ d, there are four pulsations in resonance with the orbital frequency and they all have higher amplitudes than the three current TEOs, indicating a greater resonance state when the orbital decays to $f'_{orb}$ (discussion given below). The detailed resonant frequencies with $P'_{orb}$ are shown in the last column of Table 3, and illustrated in Figure 5 by the cyan and blue lines.

Fuller [3] presented a comprehensive theoretical study to compute the resonant frequencies, amplitudes, and phases of TEOs in HB star systems. For application of this theory to an individual HB star KIC 8164262, Fuller et al. [52] determined that resonance locking can reproduce the frequency and amplitude of the strange TEO orbital frequency of $n = 229$ times. A series of studies [12,53,54] have been very successful in this regard, but the process is very complicated.

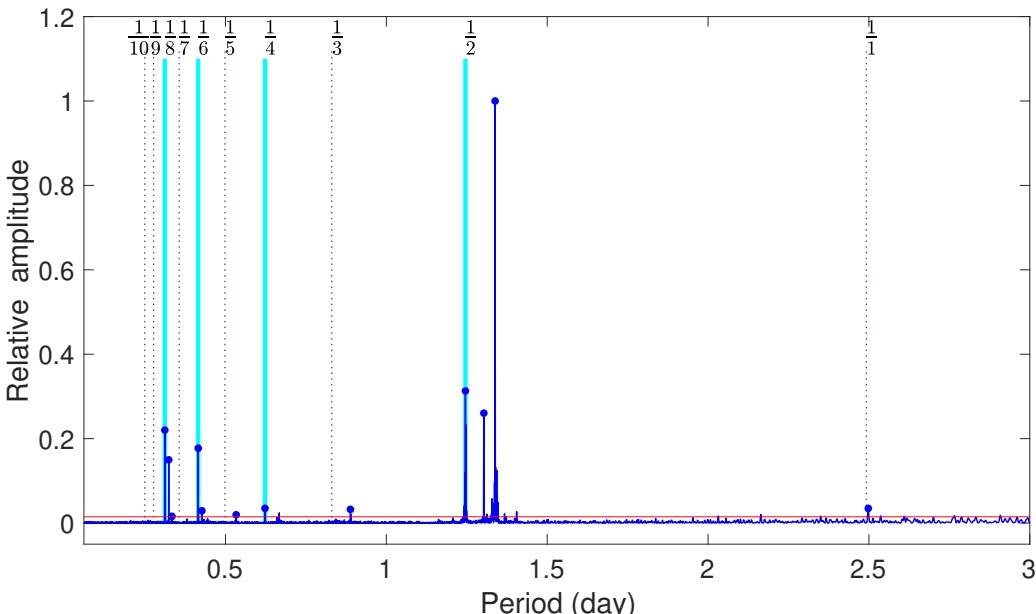

**Figure 5.** The orbital period of KIC 3766353 is predicted to decay to 2.492 days due to the arrival of the lower orbital energy configuration. The vertical blue dotted lines show the multiple periods of the decay orbit $P'_{orb}$, while the blue lines represent the stellar pulsation spectrum, which is preserved and unchanged from Figure 4. The four resonant TEOs $n' = 2, 4, 6$, and 8 have higher amplitudes than those shown in Figure 4.

For simplicity, here, we take another simple approach to demonstrate that resonant TEOs can be excited more easily when $P'_{orb} = 2.491716$ d for KIC 3766353. The Hansen coefficient $F_{nm}$ is one of the crucial factors that is used to determine the amplitudes of the TEOs [55],

$$F_{nm} = \frac{1}{\pi} \int_0^{\pi} \frac{\cos[n(E - e \sin E) - m\nu]}{(1 - e \cos E)^l} \, dE \, , \tag{7}$$

where $n$ is the multiple of orbital frequency, as mentioned in Note 1, $l$ refers to multiples of the tidal potential which excites the pulsation mode, $m$ is the azimuthal order of the spherical harmonic describing the geometry of the TEOs, $E$ is the eccentric anomaly, $\nu$ is the true anomaly, and the relation between $\nu$ and $E$ is $\tan(\nu/2) = \sqrt{(1+e)/(1-e)} \tan(E/2)$. Equation (7) is defined assuming spin–orbit alignment in the binary system.

The Hansen coefficient $F_{nm}$ describes the temporal coupling between the pulsation mode and the characteristic time of the periastron passage, such that a mode with a pulsation period closest to the characteristic time of the periastron passage should be the most strongly excited one. Therefore, using Equation (7), the amplitude of TEOs, or as a consequence, the detection ability of the modes, can be estimated when the orbital elements $e, E$, and $\nu$ from the observations and the pulsation mode properties $l$ and $m$ from the asteroseismic models are available.

We assumed that the orbital eccentricity was fixed at $e = 0.264$ and considered the TEOs potentially dominating the $l = 2$ or $l = 4$ modes. Figure 6 presents the relationship between $F_{nm}$ and the resonant number $n$ for TEOs with eight different $(l, m)$ values. For the case where the orbital period $P_{orb}$ is 2.666 days, the primary star generates three resonant frequencies of $n = 3, 5$, and 8 (cyan vertical dot-dashed line), but instead, four resonant frequencies of $n' = 2, 4, 6$, and 8 (black vertical dotted line) are generated for the case where the orbital period is reduced to $P'_{orb} = 2.491716$ d. Except for the mode of $(l = 4, m = 4)$, the coefficients $F_{nm}$ of $n' = 2, 4$ are larger than those of $n = 3, 5$ in the remainder of the seven $(l, m)$ modes. For the specific value of eccentricity, the lower orbital harmonic $n$ is expected to be observed in TEOs. Resonance $n' = 6$ only occurs when the orbital period decays to $P'_{orb}$, and a resonance of $n = 8$ occurs in both cases. In summary, in the case of

orbital decay to $P'_{\text{orb}}$, the resonant frequencies of $n' = 2, 4, 6$, and 8 have a good chance of being excited in KIC 3766353.

The decrease in the orbital period causes a coupling between the new orbital frequency and the stellar intrinsic pulsation frequency, which increases the amplitude of the stellar pulsations significantly. Such resonant states have lower orbital energy configurations; a typical example is KIC 4544587. From this perspective, KIC 3766353 would have to undergo an orbital period decrease to obtain a similar state to KIC 4544587, which means going from orbital decay to apsidal precession.

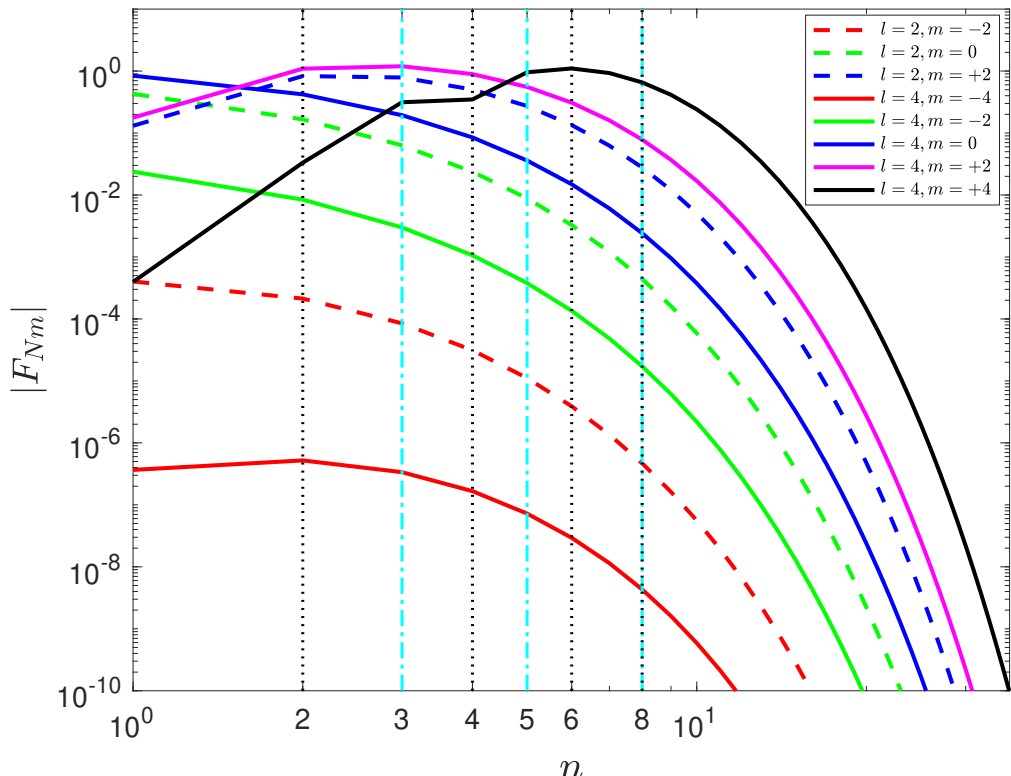

**Figure 6.** The strength of Hansen coefficients $F_{nm}$ for the KIC 3766353 system. $F_{nm}$ as a function of the orbital harmonic $n$ for different $(l, m)$ values with orbital eccentricity $e = 0.264$. The cyan vertical dot-dashed and black vertical dotted lines mark TEOs generated in the orbital periods of $P_{\text{orb}} = 2.666$ d and $P'_{\text{orb}} = 2.492$ d, respectively. See the main text for more details.

## 7. Discussion and Conclusions

Apsidal precession and orbital decay are two indistinguishable types of dynamic evolution in binary systems. Thanks to the high-precision and continuous photometric data of modern space telescopes, these two subtle types of dynamic evolution can be observed. However, these two kinds of dynamic evolution occur simultaneously in HB star systems, and their evolutionary mechanism attracts our attention. In this paper, we present detailed analyses of two eccentric binaries containing heartbeat signatures in order to investigate the correlation between stellar pulsation and orbital periods of these systems. The following conclusions can be drawn from this work.

In theory, the resonance representing the orbital local lower energy is a physical correlation between the stellar pulsation and orbital period of the HB star systems. The mass and radius of the pulsating star determine the timescale of stellar pulsation $P_{\text{pul}}$. The orbital period $P_{\text{orb}}$ is a function of the semi-major axis according to Kepler's third law. Kepler's third law acts as a bridge between the stellar pulsation period $P_{\text{pul}}$ and the orbital period $P_{\text{orb}}$. Due to tidal dissipation, the orbit of the HB star keeps shrinking, the orbital period decreases, and the orbital frequency gradually increases, constantly scanning the intrinsic frequencies of the primary star. When the ratio of $P_{\text{orb}}$ to $P_{\text{pul}}$, denoted as $P_{\text{n}}$, is

equal to an integer, the specific orbital period is captured in the resonance state, resulting in resonance locking. In the resonance state, the semi-major axis of the orbit remains constant, and the binary system only keeps the rotation of the axis connecting the periastron and apastron, i.e., apsidal precession.

In the observations, the resonance locking is well-demonstrated by the periodogram of the stellar pulsation and the orbital period of the binary system. KIC 4544587 is an apsidal precession HB star with eight observed resonant frequencies and has an almost integer ratio of $P_n = 67.968$. The resonance state with lower orbital energy keeps the HB star in a stable orbital configuration. On the contrary, KIC 3766353 currently has only three weak pulsation–orbital resonance frequencies and shows a non-integer ratio of $P_n = 83.163$, indicating that KIC 3766353 is undergoing the process of orbital shrinkage [21]. When the orbital period of KIC 3766353 decreases to 2.492 days, the system will have four higher amplitude resonant frequencies. At this time, the dynamics of KIC 3766353 will change from orbital decay to apsidal precession.

Space telescopes have observed HB stars with sufficient accuracy that longer time baselines are needed to find more sources of apsidal precession and orbital decay. Unfortunately, *TESS* has very few overlapping sources with *Kepler*. For the few sources where apsidal precession has been reported [27,56], the time baseline is too short or there are no continuous photometric data to analyze the contribution of dynamic tides to apsidal precession rates, as mentioned in Ou et al. [22]. The same obstructions occur in samples of orbital decay sources with the heartbeat signature [55,57]. Obviously, the statistical analysis will make our conclusion more convincing. However, the samples of apsidal precession and orbital decay are very small, which has prevented us from carrying out a statistical analysis with a large sample size so far.

**Author Contributions:** Conceptualization, J.-W.O. and C.Y.; methodology, J.-W.O., C.J. and M.Y.; writing—original draft preparation, J.-W.O.; writing—review and editing, J.-W.O., C.J., M.Y, D.-Y.G. and G.L.; supervision, C.Y. and G.L. All authors have read and agreed to the published version of the manuscript.

**Funding:** This work is funded by the Key Discipline Construction Project of Shaoguan University—Subject teaching (physics), the Talent Introduction Program of Shaoguan University (440-9900064601) and the Key Project of the Natural Science Research of Shaoguan University (SZ2021KJ10).

**Data Availability Statement:** The photometry data underlying this article were accessed from the Mikulski Archive for Space Telescopes (MAST): https://archive.stsci.edu/ (accessed on 15 July 2023). The derived data generated in this research will be shared on reasonable request to the corresponding author.

**Acknowledgments:** J.-W.O. is grateful to Zhu-Ji Middle School for providing a quiet research environment.

**Conflicts of Interest:** The author declares no conflict of interest.

## Note

1. The TEOs are exact integer multiples of the orbital frequency $f_{orb}$ due to additional gravity mode oscillations induced by the dynamical tide [3,4]. Throughout this paper, we refer to TEOs using their resonant numbers $n$, which are defined via their frequencies $f_{TEO} = nf_{orb}$.

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
