# Peer review of "Tidal Resonance: A Factor Worth Considering in the Orbital Evolution of Heartbeat Stars"

_universe, doi:10.3390/universe9120514_

Round 1

Reviewer 1 Report

Comments and Suggestions for Authors

This paper attempts to establish a simple relationship between stellar pulsations and orbital evolutions in heartbeat systems based on a sample of two objects. I have identified the following major points that need to be addressed, along with some minor ones.

Major points:

- As the authors mention in the second to last sentence of the conclusion section, a more statistical analysis involving a larger sample of heartbeat stars is needed in order to confirm the behaviour of heartbeat systems proposed in this study. A sample with a population of N = 2 objects is just not enough. Therefore the wording throughout the manuscript should be more cautious. Particularly, the title itself should say that this is "A **potential** simple relationship between stellar pulsations and orbital evolutions".

- Table 1 indicates that the components of KIC 4544587 are both massive A-type stars, while the components of KIC 3766353 are much cooler and less massive stars (F and K stars). Although the two systems are both heartbeat systems, the fact that they have very different fundamental stellar parameters by themselves could contribute to the different observed behaviours. This aspect should be addressed and discussed in detail either in the introduction or in the discussion section.

- The uncertainties on the computed Pn values for the two systems (Pn = 67.9683 for KIC 4544587, and Pn = 83.1630 for KIC 3766353) should be clearly quoted. This should be done by e.g. propagation of uncertainties from equation 4, given that the semi-major axis, the masses (M1,M2), and the radius R1 all have their own uncertainties (Table 1). The conclusion on whether the computed Pn is an integer or not  -- and therefore whether the orbit is stable or not -- will depend on its uncertainty. For example if it turns out that Pn = 83.1630 +- 0.1, then on can not say anything about whether the orbit is table or not.

- It is concerning that the Pn values are very high, of the order of 68 and 83 (i.e. Fpuls = 68Forb and Fpuls = 83Forb), while all the TEO frequencies reported in Tables 2 and 3 do not exceed 10Forb. That huge discrepancy is explained nowhere in the manuscript.

- Page 4, Line 114. "[...] after subtracting the orbit of binary model.". The method used to establish the binary model light curve should be described (e.g. was the binary model light curve taken from previous studies, or did the authors recalculate the binary model fit with existing codes like PHOEBE, or did the authors establish their own model with their own code?).

- Figures 2 and 3. A few points on these two figures:

-- These figures are overcrowded. The two periodograms (the one with the orbital information and the one after removal of the binary model) should be plotted in two different panels (top and bottom).

-- "the red dots mark the extracted pulsation frequencies.": how was this done? By pre-whitening? Using which code? These details should be included in the manuscript so that future studies can test the reproductibility of these results.

-- the dotted blue lines should be plotted on top of the green lines so that the latter do not cover the dotted blue lines.

- Page 9, Lines 208-209. "Assuming that the masses of the components are constant [...]". This is a strong assumption. In the case of KIC 4544587, the massive components (A-type stars) of the system have more significant mass-loss rates, meaning that the masses would decrease more over time. In this case, Kepler's third law (equation 2) implies that the orbital period should increase, not decrease.

Minor points:

- Page 1, Line 18. First sentence of the introduction. Grammar. This should be in the present tense ("brightens" instead of "brightened") since it is a general statement.

- Page 1, Line 21. Grammar. This should be "due to the companion moving close and away" instead of "due to the companion moves close and away".

- Page 1, Line 28. Grammar. "Outside solar system, heartbeat star provides [...]" should read "Outside THE solar system, heartbeat starS provide [...]".

- Page 1, Line 36. The uncertainty on the rate of of apsidal precession should also be quoted.

- Page 2, Line 52. "(i.e. the fluctuating in the light curve)". The fluctuating what? Maybe the authors meant "(i.e. the fluctuations in the light curve)"

- Figure 1. The caption should indicate by what quantity the light curves were normalized (e.g. by the median value of the out-of-eclipse variations?).

- Page 3, Line 78. "[...] been investigated in many literatures" sounds awkward. I suggest something along the lines of "[...] been investigated in many studies in the literature.".

- Page 4, Line 100 (also page 9, Line 217). "Perihelion" and "aphelion" are used for objects orbiting the Sun. Instead, "periastron" and "apastron" should be used here.

- Page 6, Line 137. "[...] does not arrive a stable state [...]" should read "[...] does not reach a stable state [...]".

- Pages 6-7, Lines 151-152. "System orbit appears to capture some stellar pulsations, [...]". It is not clear what the authors mean by that. This clause should be rephrased.

- Figures 2, 3, 4, and 5 are not color-blindness friendly (red and green should not be present together in the same plot).

- Figure 5. The caption should indicate that this is for the KIC 3766353 system.

Comments on the Quality of English Language

Moderate English revision required.

Reviewer 2 Report

Comments and Suggestions for Authors

The paper titled "A simple relationship between stellar pulsations and orbital evolutions" by Ou et al. deals with the orbital properties of two heartbeat stars, one showing evidence of apsidal precession, and one showing evidence of orbital decay.

The paper is interesting, the method of investigation fairly effective. Some clarifications and restructuring are needed to make the paper more readable. Comments relevant to science content are marked with two asterisks. And I am not sure all the conclusions might be supported by the analysis of the authors, especially about the evolution from orbital shrinking to apsidal precession. 

** Introduction: please add more information about the stars that you are considering, such a metallicity, disk/halo population, distance etc. Be more explicit about physical processes mentioned at l.22-l.23.

** The paper misses a section focused on the data and their analysis. As a consequence, it is as if a very long introduction continues down to the results obtained in this investigation. The paper should be split into introduction, data, etc. The author should stress what their contribution is. 

* l. 59: Why the orbit of KIC 4544587 is precession, in contrast, the orbit of KIC 3766353 is shrinkage. Sentence needs a rewrite. 

** Figure 2:  the black line is not visible. However, the question is: how  the false alarm probability was computed at 0.1%? 

** I am not sure the present analysis provides convincing results especially about the evolution. The authors  should elaborate on the the physical mechanism that can yield the transition from shrinking to resonance  locking.   References to simulations could be helpful. 

** l. 145: "Assuming that the properties of the primary star remain constant, that is, the eigenfrequencies of the stellar pulsations do not vary with time. " This sentence  should be rewritten. However, the assumption is likely an oversimplification, as orbital changes are induced by exchange of matter and momentum  at periastron. The authors should be more explicit on what is happening at periastron.  Note also that the frequency of star oscillation reported in Eq. 3  is due to intrinsic oscillation modes, not necessarily tidally-induced.  

Comments on the Quality of English Language

English is almost fine, some corrections needed. 
